# Experimental Results of Partial Discharge Localization in Bounded Domains

**DOI:** 10.3390/s21030935

**Published:** 2021-01-30

**Authors:** Luca Perfetto, Gabriele D’Antona

**Affiliations:** Department of Energy, Politecnico di Milano, 20156 Milan, Italy; gabriele.dantona@polimi.it

**Keywords:** partial discharges, antenna, localization, eigenfunctions, IEC TS 62478:2016

## Abstract

This work presents a novel diagnostic method to localize Partial Discharges (PDs) inside Medium Voltage (MV) and High Voltage (HV) equipment. The method is well suited for that equipment presenting a bounded domain with fixed Boundary Conditions (BCs) such as Oil-Filled Power Transformers (OFPTs), Air Insulated Switchgears (AISs), Gas Insulated Switchgears (GISs) or Gas Insulated Transmission Lines (GILs). It is based on Electromagnetic (EM) measurements which are used to reconstruct the EM field produced by the PD and localize the PD itself. The reconstruction and localization tasks are based on the eigenfunctions series expansion method which intrinsically accounts for the physical information of the propagation phenomenon. This fact makes the proposed diagnostic method very robust and accurate even in real and complex scenarios. The promising experimental results, obtained in two different test cases, confirmed the ability and powerfulness of the proposed PD localization method.

## 1. Introduction

Nowadays, especially in Medium Voltage (MV) and High Voltage (HV) systems, the electrical insulation of each apparatus is a critical aspect to be monitored. In presence of an improper components installation or material manufacturing defects, insulation parts are subjected to an high and inhomogeneous electric field which causes Partial Discharges (PDs) [1]. A PD consists of a displacement of electric charges which is responsible for a localized thermal dissipation that implies a progressive deterioration of dielectric components eventually leading to a breakdowns [1,2,3]. Hence, in order to ensure the electrical systems safety and reliability and, in order to prevent catastrophic failures and consequent high outage costs, an effective diagnostic method is necessary to detect and localize PDs.

At present, PD diagnostics is applied to power cables, rotating machines, electrical switchgears (i.e., air or gas insulated), power transformers (i.e., resin dry-type or oil-filled) and insulated transmission lines. It mainly consists of signal acquisition and subsequent signal processing [4,5,6,7,8]. PD signal acquisition is in general difficult to perform since the discharge channel is very narrow (with a radius of hundred of micrometers), the phenomenon evolves in a very short time (tens of ns) and it presents a limited small amount of emitted energy.

PDs measurements are carried out using the conventional approach according with IEC 60270 [9] (i.e., electric current and voltage measurements) or the unconventional approach according with IEC 62478 [10] (i.e., electromagnetic and acoustic measurements). This latter is suitable to both PD detection and localization, which is carried out using the Time Difference Of Arrival (TDOA) [11,12,13] or the Received Signal Strength (RSS) measurements [14,15]:TDOA is the time required by the signal to propagate from a reference receiver, that is, ${\vec{r}}_{1}$, to the *i*-th receiver ${\vec{r}}_{i}$. Mathematically, this concept is expressed with the measurement model
(1)$${∥{\vec{r}}_{s}-{\vec{r}}_{i}∥}_{2}-{∥{\vec{r}}_{s}-{\vec{r}}_{1}∥}_{2}=v\left(t_{i}-t_{1}\right),\;\;i\neq \;1,$$
where ${\vec{r}}_{s}$ is the source location, ${∥{\vec{r}}_{s}-{\vec{r}}_{i}∥}_{2}$ is the distance between the source and the *i*-th receiver, while $t_{i}$ is the signal arrival time in ${\vec{r}}_{i}$.the RSS is based on the average EM radiation power emitted by the source and detected at receivers locations. The RSS measurement model for a single receiver is [16,17]
(2)$$P_{i}^{r}=K_{p}\frac{P^{t}}{{∥{\vec{r}}_{s}-{\vec{r}}_{i}∥}_{2}^{\gamma }},$$
where $P_{i}^{r}$ is the average power measured by the i-th receiver, $P^{t}$ is the average power transmitted by the source, γ is the path loss exponent (in free space: γ=2) and $K_{p}$ is a constant. In PD localization applications, terms $P^{t}$ and $K_{p}$ in (Equation 2) are unknown. In order to overcome this issue, it is sufficient to set a reference receiver (i.e., located in ${\vec{r}}_{1}$). In this way, the average received power in ${\vec{r}}_{i}$ is related to the received power in ${\vec{r}}_{1}$ according with
(3)$$P_{i}^{r}=P_{1}^{r}{\left(\frac{{∥{\vec{r}}_{s}-{\vec{r}}_{1}∥}_{2}}{{∥{\vec{r}}_{s}-{\vec{r}}_{i}∥}_{2}}\right)}^{\gamma },$$
where $P_{1}^{r}$ is the average power measured by the reference receiver.

However, both TDOA and RSS methods presents some issues. For the purpose of computing the *N* unknown source coordinates (vector ${\vec{r}}_{s}$), a set of M>N non-linear equations has to be solved, where *M* is the number of receivers. Furthermore, localization accuracy is limited by the presence of several error sources, such as the intrinsic noise of measurement hardware, or impulsive and periodic external interferences of abrupt switching operations or radio and TV broadcastings [18]. Moreover, in (Equation 1), the propagation speed *v* is not perfectly known and constant everywhere as often assumed [19,20,21]. Equation (Equation 3) is only valid for distances located in the far field region while the path loss exponent γ is environment-dependent and it is assumed constant everywhere in the propagation domain [22]. For all these reasons, TDOA and RSS methods are not able to always accurately localize PDs in finite and closed domains such as Electrical Switchgears (SW), Oil-Filled Power Transformers (OFPT), Gas-Insulated Switchgears (GIS) or Gas-Insulated Transmission Lines (GITL). These are MV and HV components, which volume is bounded by metallic surfaces and may contain several insulation parts manifesting PDs. Inside these domains, EM wave propagation is very complex since it is affected by reflection, refraction and diffraction effects [23].

In this work, the PD localization is performed exploiting the novel method based on EM measurements well described in [24,25,26]. Its core is represented by the eigenfunctions expansion method which accurately describe the EM wave field propagation inside bounded domains. This mathematical tools, is used as the basis for the Inverse Problem (IP) solution aimed to reconstruct the EM wave field inside the domain and to localize the PD source. The paper is organized in the following way—the formalized EM model and numerical algorithm are presented in Section 2. The experimental setup used to validate the method is described in Section 3. In Section 4 the results are presented for two different test cases. The conclusion is in Section 5.

## 2. Method

The EM wave propagation problem due to a PD can be formalized as [27]
(4)$$\left\{\begin{array}{c} L_{E}\vec{E}\left(\vec{r},t\right)+\frac{\partial^{2}}{\partial t^{2}}\underline{I}\;\cdot \;\vec{E}\left(\vec{r},t\right)=\vec{f_{E}}\left(\vec{r},t\right)\\ L_{H}\vec{H}\left(\vec{r},t\right)+\frac{\partial^{2}}{\partial t^{2}}\underline{I}\;\cdot \;\vec{H}\left(\vec{r},t\right)=\vec{f_{H}}\left(\vec{r},t\right),\\ \mathrm{BCs}:arbitrary\\ \mathrm{ICs}:arbitrary \end{array}\right.$$
where

$\vec{E}\left(\vec{r},t\right)$ and $\vec{H}\left(\vec{r},t\right)$ are the electric and magnetic field vectors;$\vec{f_{E}}\left(\vec{r},t\right)$ and $\vec{f_{H}}\left(\vec{r},t\right)$ are the electric and magnetic source vectors;$\underline{I}$ is the unit dyadic;BCs and ICs are the Boundary and Initial Conditions;the quantities:
(5)$$\left\{\begin{array}{c} L_{E}\left(\;\cdot \;\right)={\underline{\epsilon }}^{-1}\;\cdot \;\nabla \times {\underline{\mu }}^{-1}\;\cdot \;\nabla \times \left(\;\cdot \;\right)\\ L_{H}\left(\;\cdot \;\right)={\underline{\mu }}^{-1}\;\cdot \;\nabla \times {\underline{\epsilon }}^{-1}\;\cdot \;\nabla \times \left(\;\cdot \;\right) \end{array}\right.$$
are the linear spatial operators respectively for the electric and magnetic field. In (Equation 5) the magnetic permeability $\underline{\mu }=\underline{\mu }\left(\vec{r}\right)$ and the electric permittivity $\underline{\epsilon }=\underline{\epsilon }\left(\vec{r}\right)$ are in general tensor quantities.

In presence of fixed and homogeneous BCs, solution to (Equation 4) is [27]
(6)$$\left\{\begin{array}{c} \vec{E}\left(\vec{r},t\right)=\sum_{k=1}^{\infty }{\vec{\phi }}_{k}\left(\vec{r},t\right)\;\cdot \;m_{k}\\ \vec{H}\left(\vec{r},t\right)=\sum_{k=1}^{\infty }{\vec{\psi }}_{k}\left(\vec{r},t\right)\;\cdot \;n_{k}, \end{array}\right.$$
where $m_{k}$ and $n_{k}$ are dimensionless weighting coefficients determined by the electric and magnetic source vectors and prescribed ICs, while ${\vec{\phi }}_{k}={\vec{\phi }}_{k}\left(\vec{r},t\right)$ and ${\vec{\psi }}_{k}={\vec{\psi }}_{k}\left(\vec{r},t\right)$ are the problem eigenfunctions, which can be can be factorized as [28]
(7)$$\left\{\begin{array}{c} {\vec{\phi }}_{k}\left(\vec{r},t\right)={\vec{e}}_{k}\left(\vec{r}\right)v_{k}\left(t\right)\\ {\vec{\psi }}_{k}\left(\vec{r},t\right)={\vec{h}}_{k}\left(\vec{r}\right)w_{k}\left(t\right), \end{array}\right.$$
where ${\vec{e}}_{k}\left(\vec{r}\right)$, ${\vec{h}}_{k}\left(\vec{r}\right)$ and $v_{k}\left(t\right)$, $w_{k}\left(t\right)$ are solutions of two eigenvalues problems with eigenvalues $\lambda_{ek}^{2}$ and $\lambda_{hk}^{2}$, to which correspond the eigenfrequencies $f_{ek}=\frac{\lambda_{ek}}{2\pi }$ and $f_{hk}=\frac{\lambda_{hk}}{2\pi }$ [28].

Practical implementation of infinite series expansions in (Equation 6) is not feasible and these summations are limited to the first *N* eigenfunctions. Therefore, also according to (Equation 7), (Equation 6) it is rewritten as
(8)$$\left\{\begin{array}{c} \vec{E}\left(\vec{r},t\right)=\sum_{k=1}^{N}m_{k}\;\cdot \;{\vec{e}}_{k}\left(\vec{r}\right)v_{k}\left(t\right)\\ \vec{H}\left(\vec{r},t\right)=\sum_{k=1}^{N}n_{k}\;\cdot \;{\vec{h}}_{k}\left(\vec{r}\right)w_{k}\left(t\right). \end{array}\right.$$

Let us consider the electric field expansion in (Equation 8) only for the sake of brevity (it is valid the dual for the magnetic field $\vec{H}$), the *i*-th spatial component of the measured electric field $E^{i}$ by a antenna in $\vec{r_{j}}$ at time $t_{q}$ is
(9)$$E^{i}\left({\vec{r}}_{j},t_{q}\right)=\sum_{k=1}^{N}m_{k}\;\cdot \;e_{k}^{i}\left({\vec{r}}_{j}\right)v_{k}\left(t_{q}\right).$$

Defining a set of *M* antennas and *P* sampling time instants, (Equation 9) can be expressed as
(10)d=Am
where
(11)$$\underset{(M\times P)\times 1}{d}=\left[\begin{array}{c} E^{i}\left({\vec{r}}_{1},t_{1}\right)\\ \vdots \\ E^{i}\left({\vec{r}}_{M},t_{1}\right)\\ E^{i}\left({\vec{r}}_{1},t_{2}\right)\\ \vdots \\ E^{i}\left({\vec{r}}_{M},t_{P}\right). \end{array}\right]$$
(12)$$\underset{(M\times P)\times N}{A}=\left[\begin{array}{cc} e_{1}^{i}\left({\vec{r}}_{1}\right)v_{1}\left(t_{1}\right)\cdots & e_{N}^{i}\left({\vec{r}}_{1}\right)v_{N}\left(t_{1}\right)\\ \vdots & \vdots \\ e_{1}^{i}\left({\vec{r}}_{M}\right)v_{1}\left(t_{P}\right)\cdots & e_{N}^{i}\left({\vec{r}}_{M}\right)v_{N}\left(t_{P}\right)\\ e_{1}^{i}\left({\vec{r}}_{M}\right)v_{1}\left(t_{1}\right)\cdots & e_{N}^{i}\left({\vec{r}}_{M}\right)v_{N}\left(t_{1}\right)\\ \vdots & \vdots \\ e_{1}^{i}\left({\vec{r}}_{M}\right)v_{1}\left(t_{P}\right)\cdots & e_{N}^{i}\left({\vec{r}}_{M}\right)v_{N}\left(t_{P}\right) \end{array}\right],\underset{N\times 1}{m}=\left[\begin{array}{c} m_{1}\\ \vdots \\ m_{N} \end{array}\right]$$

Once the electric field measurements and the eigenfunctions are available, the amplitude coefficient vector m has to be determined solving the IP [29]
(13)$$\tilde{m}=A^{†}d,$$
where
(14)$$A^{†}={\left(A^{T}A+\xi I\right)}^{-1}A^{T}$$
is the pseudo-inverse matrix of the problem, ξ is the Tikhonov regularization parameter [30] and I is the identity matrix. The electric field distribution is subsequently reconstructed according with
(15)$${\tilde{E}}^{i}\left(\vec{r},t\right)=\sum_{k=1}^{N}{\tilde{m}}_{k}\;\cdot \;e_{k}^{i}\left(\vec{r}\right)v_{k}\left(t\right).$$

Back-propagating in time the reconstructed electric field distribution, the PD sources are localized in those volumes presenting the highest EM energy density.

### 2.1. Ill-Posedness and Regularization

The regularization parameter ξ is introduced since the IP (Equation 13) is ill-posed [31]. In order to get a physical solution, its value needs to be chosen according with the available information on the specific physical model and noise level. The employed method in this work is performed using *a posteriori* criterion based on the L-curve [32,33]. It is a parametric plot of the misfit and stabilizing functionals both depending on the regularization parameter. The resulting curve presents a corner located in correspondence of the optimal regularization parameter, which value is a trade-off between the best fitting and most reasonable stabilization.

### 2.2. Localization Accuracy

A suitable error should be defined in order to evaluate the accuracy of the IP solution and the reconstructed field quality. The error definition formalized here is based on common estimation theory concepts.

For a chosen time instant $t=t^{*}$, the reconstructed field spatial energy density (electric or magnetic) $w(r_{s},t=t^{*})$ is assumed to be proportional to the probability density distribution of the PD position, here represented by the random column vector $r_{s}$ (column vectors $r_{s}$ are used for the sake of simplicity in this formalization. They should not be confused with geometric ones $\vec{r}$.). The mean vector ${\tilde{r}}_{s}$ and the covariance matrix Σ of $r_{s}$ are respectively [34]
(16)$$\underset{\left(L\times 1\right)}{{\tilde{r}}_{s}}=\frac{\int_{\Omega }r_{s}\;\cdot \;w\left(r_{s},t=t^{*}\right)\;dr_{s}}{\int_{\Omega }w\left(r_{s},t=t^{*}\right)\;dr_{s}},$$
(17)$$\begin{array}{c} \underset{\left(L\times L\right)}{\Sigma }=\frac{\int_{\Omega }\left(r_{s}-{\tilde{r}}_{s}\right){\left(r_{s}-{\tilde{r}}_{s}\right)}^{T}\;\cdot \;w\left(r_{s},t=t^{*}\right)dr_{s}}{\int_{\Omega }w\left(r_{s},t=t^{*}\right)\;dr_{s}}. \end{array}$$

The localization error vector $\Delta r_{s}$ is
(18)$$\underset{\left(L\times 1\right)}{\Delta r_{s}}=r_{s}-{\bar{r}}_{s},$$
which can be rewritten as
(19)$$\Delta r_{s}=r_{s}-{\bar{r}}_{s}+{\tilde{r}}_{s}-{\tilde{r}}_{s}=\left(r_{s}-{\tilde{r}}_{s}\right)+b,$$
where $\underset{\left(L\times 1\right)}{b}={\tilde{r}}_{s}-{\bar{r}}_{s}$ is the estimator bias vector. An estimation of the localization accuracy can be retrieved computing the Euclidean norm squared of (Equation 19) which leads to
(20)$$\begin{array}{cc} \; & E\left\{\Delta {r_{s}}^{T}\Delta r_{s}\right\}=\\ & =E\left\{{\left(r_{s}-{\tilde{r}}_{s}\right)}^{T}\left(r_{s}-{\tilde{r}}_{s}\right)\right\}+E\left\{b^{T}b\right\}+2E\left\{{\left(r_{s}-{\tilde{r}}_{s}\right)}^{T}b\right\}=\\ \; & =tr\left(\Sigma \right)+{∥b∥}_{2}^{2}+2E\left\{{\left(r_{s}-{\tilde{r}}_{s}\right)}^{T}b\right\}, \end{array}$$
where $E\left\{\;\cdot \;\right\}$ is the expected value operator and $tr\left\{\;\cdot \;\right\}$ the trace operator. Since in (Equation 20) the last term on the RHS is
(21)$$\begin{array}{cc} \; & E\left\{{\left(r_{s}-{\tilde{r}}_{s}\right)}^{T}b\right\}=\\ \; & =E\left\{r_{s}^{T}\right\}{\tilde{r}}_{s}-E\left\{r_{s}^{T}\right\}{\bar{r}}_{s}-{\tilde{r}}_{s}^{T}{\tilde{r}}_{s}+E\left\{{\tilde{r}}_{s}^{T}\right\}{\bar{r}}_{s}=\\ \; & ={\tilde{r}}_{s}^{T}{\tilde{r}}_{s}-{\tilde{r}}_{s}^{T}{\bar{r}}_{s}-{\tilde{r}}_{s}^{T}{\tilde{r}}_{s}+{\tilde{r}}_{s}^{T}{\bar{r}}_{s}=0, \end{array}$$
(Equation 20) is the estimator Mean Squared Error (MSE) [34]
(22)$$\begin{array}{cc} MSE=E\left\{\Delta {r_{s}}^{T}\Delta r_{s}\right\} & =tr\left(\Sigma \right)+{∥b∥}_{2}^{2}, \end{array}$$
from which the Root Mean Squared Error (RMSE) is
(23)$$\begin{array}{c} RMSE=\sqrt{MSE}=\sqrt{tr\left(\Sigma \right)+{∥b∥}_{2}^{2}}. \end{array}$$

The MSE (or the RMSE) is computed at each field reconstruction time step and it presents a minimum when the reconstructed field distribution converges toward the PD location and its nearby surrounding volume includes the highest EM energy density. Therefore, (Equation 22) (or (Equation 23)) can be used to identify the PD source and to estimate the localization accuracy.

## 3. Experimental Setup

The test circuit used for the experimental validation is shown in Figure 1. It is composed by two main parts carrying out PD generation and detection, which are both described in the two next Section 3.1 and Section 3.2.

### 3.1. Partial Discharge Generation

The PD generation part includes:a 100 V / 30 kV MV Voltage Transformer (VT), which low voltage terminal is fed by a 50Hz source operating in the range 0V÷100V;a 100 MΩ resistor damping R in order to limit the output current in case of total discharge;a PD Generator (PDG) custom designed and realized to generate artificial PDs such as Corona, internal or surface discharges in a well defined volume. It allows the electric stress control according to different electrodes shapes and distances (d${}_{1}$ and d${}_{2}$) and eventually with the usage of different dielectric media (Figure 2). The PDG characteristics are reported in Figure 2b.a MV power cable connecting the VT and PDG. The cable is a single phase, custom designed by EssexX−RayLtd. with double shields and rated voltage of 45 kV AC. The cable termination is ad-hoc realized in order to reduce the electric stress and limit the unwanted PD activity. The cable cross section is shown in Figure 3, while the cable mechanical details are reported in Figure 3b.a metal-enclosure emulates the bounded domain under test. Its internal volume contains the MV power supply cable and the PDG. The enclosure is a 1m×1m×1m, made of Aluminum and filled by air at room temperature, pressure and humidity. The cable entry is located exactly above the PDG, ensuring an adequate distance from the enclosure surfaces, in order to avoid fault events. Figure 4 shows the internal metal-enclosure volume.

### 3.2. Partial Discharge Detection

The PD detection part refers to the measurement system employed for EM radiation measurement generated by the PD. It includes: a Data AcQuisition (DAQ) system, Transmission Lines (TLs) and a set of EM probes.

The DAQ collects simultaneously the EM signals coming from the probes. It is the Rohde&SchwarzGmbH oscilloscope RTO 2014 having 1 GHz analog frequency BW (3 dB), 4 simultaneous channels and a maximum sampling frequency of 10 GHz [35]. A set of TLs, made by 1.2
m coaxial cables RG-58, connects the probes to the oscilloscope.

The EM probe locations and physical dimensions are mainly imposed by safety reasons and measurements requirements (i.e., frequency bandwidth, sensitivity, spatial resolution). The adopted probes are cylindrical monopole antennas 50 mm long. The monopole antenna is a quarter-wavelength configuration and in an homogeneous and free domain (with ϵ=$\epsilon_{0}$ and μ=$\mu_{0}$) resonates at $f_{r}≃1.5G\mathrm{Hz}$ [36]. This kind of antennas are often used for PD measurements since they are simple and small enough to be easily manufactured [37,38,39,40]. For this specific application, they are installed inside the enclosure, far away from the live components and they are connected to SMA panel connectors directly fixed on the inner part of the enclosure surfaces, which behaves as a ground plane (Figure 4 and Figure 5a). The Return Loss (RL) ${\bar{S}}_{11}\left(f\right)$ of the used monopole antenna on a circular ground plane (diameter of ∅ 300 mm) is measured using the Vector Network Analyzer Keysight Technologies E5071C [41]. The RL magnitude $|{\bar{S}}_{11}\left(f\right)|$ and phase shift $∠{\bar{S}}_{11}\left(f\right)$ are shown in Figure 5b. Up to the resonance at ${\tilde{f}}_{r}≃1.1G\mathrm{Hz}$, the magnitude is constant while the phase shift is linear. Fixing as the maximum reconstructed eigenfrequency as $f_{eN}=0.7\;G\mathrm{Hz}$, there is no need of an antenna transfer function compensation. Although the antenna sensitivity is low, the results obtained in [24,25,26] showed that the localization accuracy is order of centimeters, compliant with the components dimensions installed inside real apparatus (i.e., support insulators in switchgears).

Figure 6 shows the measurement system composition (Probes, DAQ and TL).

### 3.3. Noise Sources

The acquired on-field PD measurements are subjected to inevitable noise due to [18,42,43,44]:measurement acquisition system and environment;communication systems (i.e., mobile communication), radio and TV broadcasting;periodic switching operations (i.e., power electronics valves commutations);stochastic events (i.e., lightning, circuit breaker trips).

Therefore, suitable de-noising techniques implementation is necessary before any features extraction from detected PD signals. However, it should be noted that the used DUT is composed by a metal-enclosure, which behaves as an effective EM shield and strongly limited external noise sources. This condition is also verified in closed equipments such as SWs, OFPTs, GILs. Furthermore, the used power cable is PD free; hence, the measurement acquisition system noise is expected only and it is modelled as random White Noise (WN).

For the experimental noise characterization, the MV cable is supplied with 30 kV without any connection to the PDG. A 50 mm monopole antenna is installed inside the DUT and a set of EM measurements are acquired in observation time lasting 100 ns at a sampling frequency of 10 GHz. The background noise Power Spectral Density (PSD) is computed with the Bartlett’s method [45] using 50 consecutive acquisitions and it is shown in Figure 7. As expected, it is a random Gaussian WN with a PSD of $≃-85{}_{V{}^{2}{\mathrm{Hz}}^{-1}}$.

### 3.4. Electromagnetic Model

According to the EM model described in Section 2, a discrete frequencies (or eigenfrequencies) spectrum is expected. In order to verify this hypothesis, MV power cable is connected to the PDG and supplied with 30 kV. A 50 mm monopole antenna is installed inside the DUT and several EM field measurements are accomplished adopting an observation time window lasting 100 ns at a sampling frequency of 10 GHz. The signal PSD is computed with the Bartlett’s method [45] using 50 consecutive acquisitions and it is shown in Figure 8.

Moreover, in Figure 8, the signal peaks are marked according with their closest numerical eigenfrequencies. The mismatch Δf between the numerical and experimental eigenfrequencies is defined as
(24)$$\Delta f=\frac{f-f_{ek}}{f_{ek}},$$
where *f* and $f_{ek}$ are the experimental and numerical frequencies respectively. Figure 9 shows the evaluated mismatch; hence, the domain eigenfrequencies are identified with an error less than 5%. Slight differences are present and they are mainly due to the uncertainty on the domain dimensions and to the PSD frequency resolution.

## 4. Results

In the following sections, two different case studies used to test the localization algorithm are presented. The chosen cases studies are demonstrative and they include some practical aspects, such as complex geometry and inhomogeneities. The numerical eigenfunctions computation is done only once and before the PD measurements (off-line) by means of the commercial software COMSOL Multiphysics^®^ [46]. By a computer having a CPU @ 1424 × Intel ® Xeon *®* E5-2697 v3 (14 cores, 2.6 GHz) and a RAM @ 64 GB the computational time is of the order of few hours. The PD localization algorithm is implemented in Matlab^®^ [47]. In less than 2 min, the latter is able to load the available pre-computed eigenfunctions, the EM measurements and to reconstruct the EM field in order to locate the PD source. To begin with, the number of processed antennas is M=4 and the sampling frequency is $f_{s}=10\;G\mathrm{Hz}$.

### 4.1. Test Case 1

In this first test case, the 3D domain contains the PDG and power supply cable only (Figure 10). The PDG is here identified as a point source located on the top part of the power cable (blue dot *s* in Figure 10). The antennas are installed inside the domain, on one single boundary surface, far away from both the PD source and power supply cable (red dots *p${}_{i}$*
Figure 10). Their exact locations are listed in Table 1.

The electric field eigenfunctions used by the algorithm are numerically computed; the first and highest order eigenfunctions are shown in Figure 11. As the frequency increases, the oscillating nature increases as well and consequently the algorithm spatial resolution improves.

The regularization parameter ξ is chosen according with the L-curve method: the IP is solved several times varying ξ only, using the same data set (d) and freezing all the others parameters (*M*,*N*,*P*). The estimated optimal value for the regularization parameter is $\xi_{opt}=0.1$ and it corresponds to the corner of the curve shown in Figure 12.

The detected electric field measurements are shown in Figure 13. According with these signals, the PD source starting time is expected to be around 28 ns; before this latter, only noise is present (noise floor < 1 mV).

Figure 14 shows the reconstructed electric field distribution, which back-propagated in time, tends to converge towards the PD source location at t≃28.1 ns. In this time instant the RMSE function which presents a minimum at the same time instant with a value of ten of centimeters (Figure 15). The other minimum at t≃29.7ns is not considered since it is not related to the starting time of the EM wave propagation (Figure 13). The proposed method successfully localize the PD source. The reduction of processed antenna signals (*M*) and sampling frequency ($f_{s}$) result in signal-to-noise ratio rising, IP algorithm simplification (i.e., less samples to process) and hardware cost reduction.

### 4.2. Test Case 2

In this test case, dielectric and ferromagnetic parts are located inside the 3D domain which presences cause reflection, refraction and diffraction of the EM wave. The dielectric part is an hollow cylinder made of Polyoxymethylene, while the ferromagnetic part is a rectangular parallelepiped made of Iron. These parts are not connected to the power supply cable and they are far away from the PD source.

The PD source is still located on the top part of the power cable (blue dot *s* in Figure 16), while the antennas locations (red dots *p${}_{i}$*
Figure 16) are the same of test case 1 (Table 1).

The electric field eigenfunctions used by the algorithm are numerically computed; the first and highest order eigenfunctions are shown in Figure 17. In order to ensure the same spatial resolution of Section 4.1, an higher number of eigenfunctions is used because of the increased volume complexity. Moreover, due to the high oscillating nature of the highest eigenfunction order, a larger regularization parameter value is chosen ($\xi_{opt}=1$) using the same approach of Section 4.1.

The detected electric field measurements are shown in Figure 18. According with these signals, the PD source starting time is expected to be around 29 ns; before this latter, only noise is present (noise floor < 1 mV).

Figure 19 shows the reconstructed electric field distribution, which back-propagated in time, tends to converge towards the PD source location at t≃29 ns. In this time instant the RMSE function which presents a minimum at the same time instant with a value of ten of centimeters (Figure 20). The proposed method successfully localize the PD source also in this more complex scenario.

## 5. Conclusions

This paper deal with novel diagnostic method for PD localization inside complex bounded domains based on EM measurements. These latter can be any closed electrical equipment presenting fixed BCs such as OFPT, AIS, GIS, GIL.

The proposed diagnostic method is based on a eigenfunctions, which intrinsically account for the physical information about the EM propagation phenomenon. Moreover, the eigenfunctions can be numerically computed for any spatial domain using numerical tools, always before the PD measurements. This fact enhances the diagnostic method applicability on-line and in complex electrical equipments including anisotropies and inhomogeneities.

In order to test the performance of the PD localization algorithm, two test cases were chosen as demonstrative while keeping practical complexities. The obtained results shown the efficacy and robustness of the proposed PD localization method. Further activities will require ad-hoc designed antennas, the formalization of an optimal strategy for the algorithm parameters choice and for the antenna location also considering real constrains of the equipment under test.

## Figures and Tables

**Figure 1 sensors-21-00935-f001:**
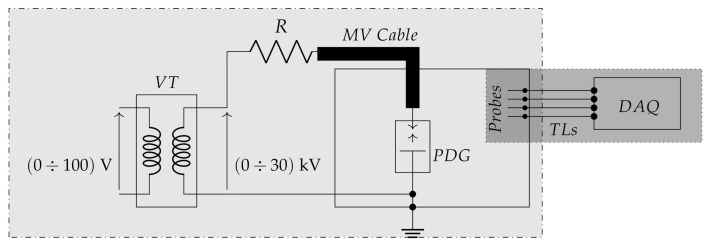
Experimental setup. VT: Voltage Transformer; R: Resistor; PDG: Partial Discharge Generator; TLs: Transmission Lines; DAQ: Data AcQuisition. Dash-dotted rectangle (light gray): PD generation. Dotted rectangle (darker gray): PD detection.

**Figure 2 sensors-21-00935-f002:**
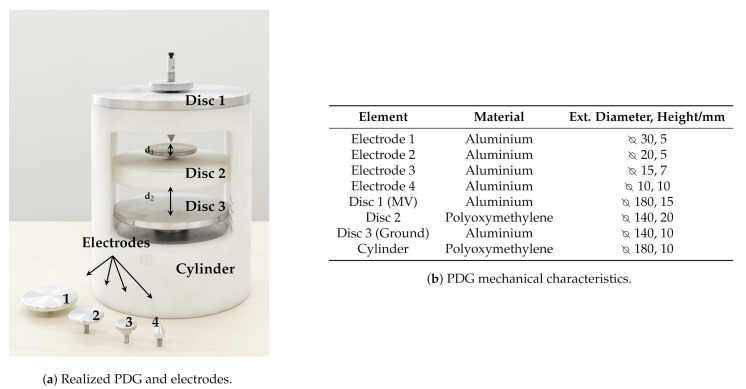
Partial Discharge Generator (PDG).

**Figure 3 sensors-21-00935-f003:**
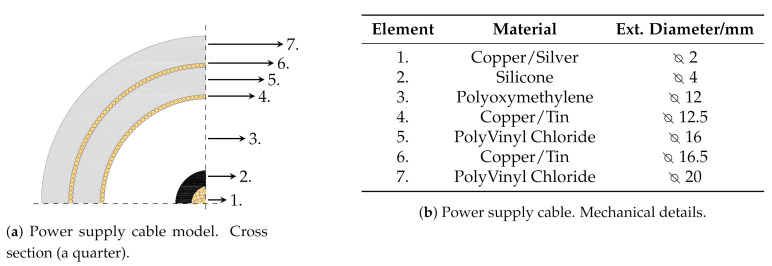
Power supply cable.

**Figure 4 sensors-21-00935-f004:**
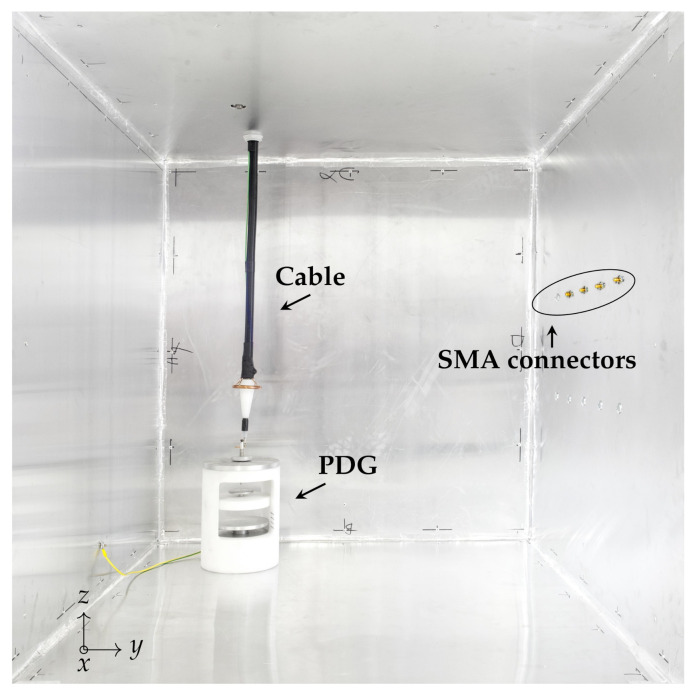
PD generation. The PDG, cable and SMA connectors (correspondent to the probes locations).

**Figure 5 sensors-21-00935-f005:**
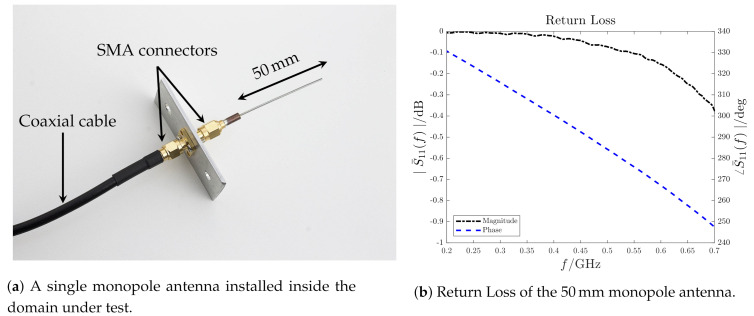
Monopole antenna.

**Figure 6 sensors-21-00935-f006:**
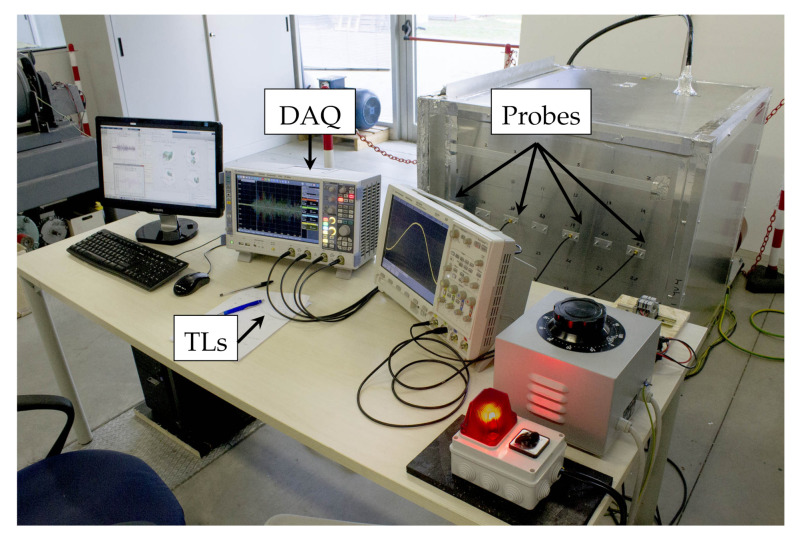
PD detection. Measurement system: probes, DAQ and TLs.

**Figure 7 sensors-21-00935-f007:**
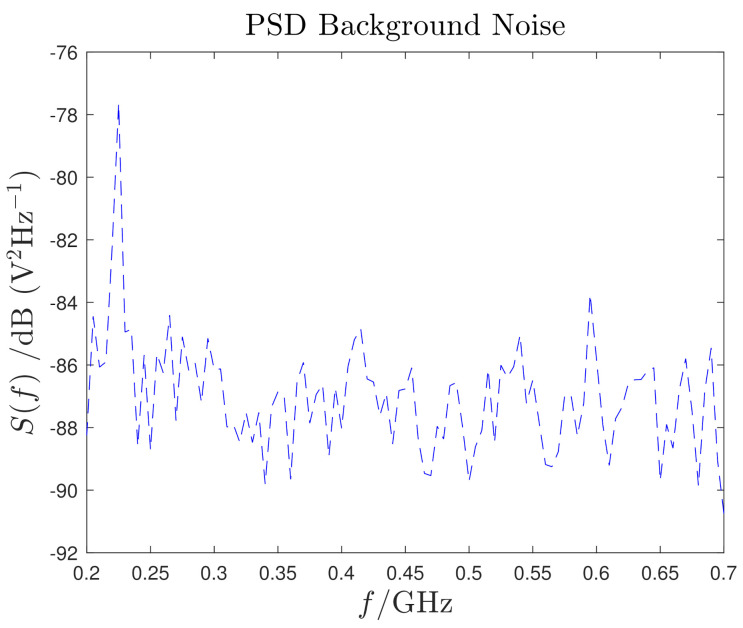
Background noise PSD.

**Figure 8 sensors-21-00935-f008:**
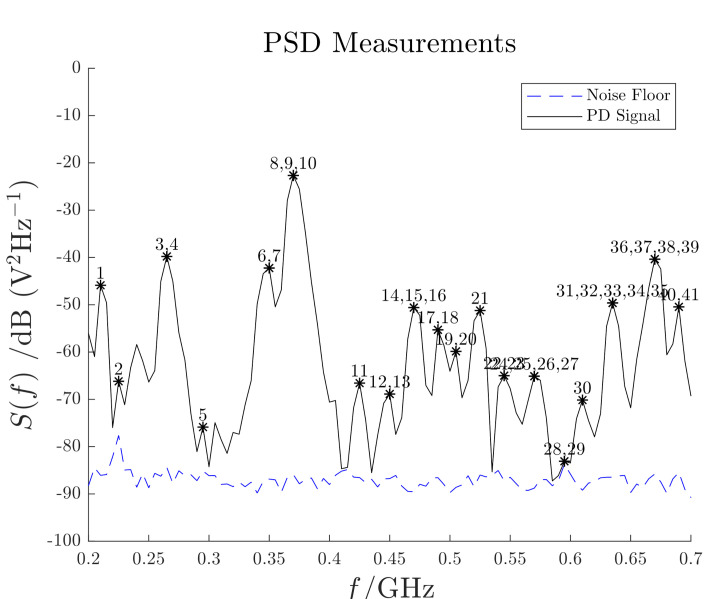
PD signal PSD. Each number corresponds to the closest numerical eigenfrequency. The dashed line shows the background noise of Figure 7.

**Figure 9 sensors-21-00935-f009:**
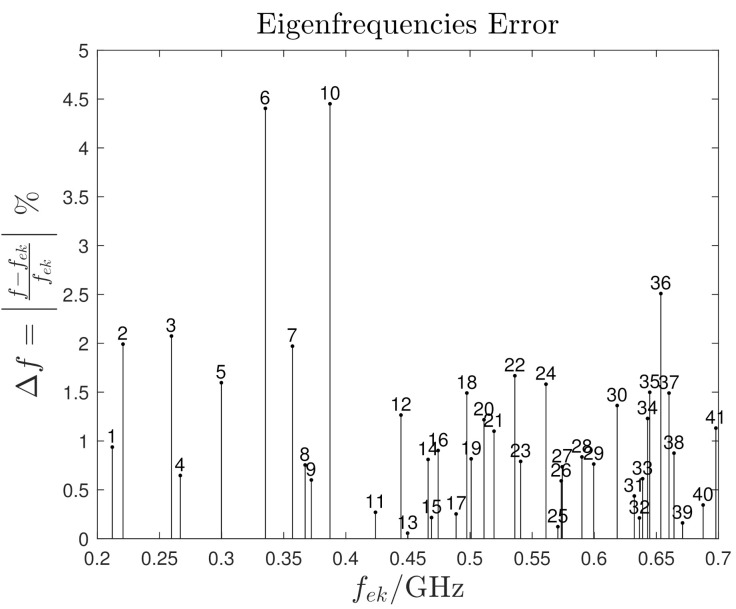
Eigenfrequencies error. Each stem corresponds to the numerical eigenfrequency $f_{ek}$.

**Figure 10 sensors-21-00935-f010:**
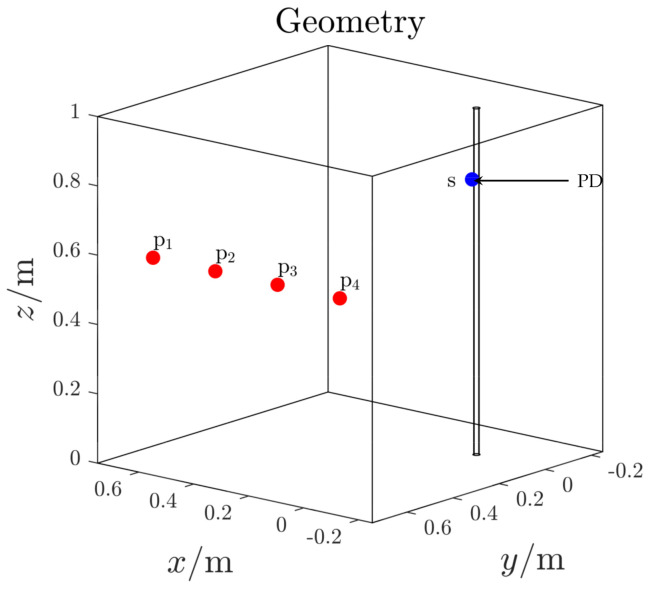
Test case 1. 3D domain.

**Figure 11 sensors-21-00935-f011:**
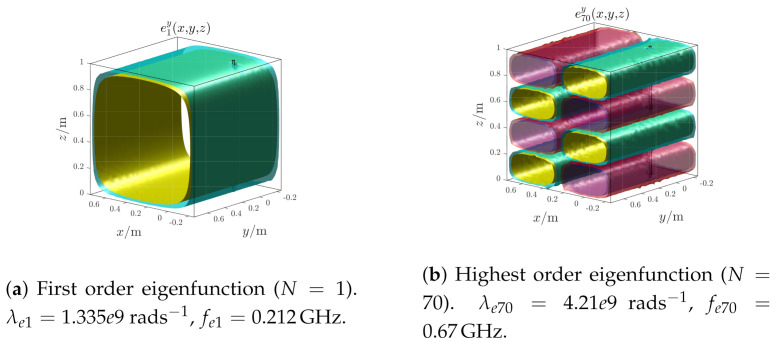
Test case 1. First and highest order electric field eigenfunctions.

**Figure 12 sensors-21-00935-f012:**
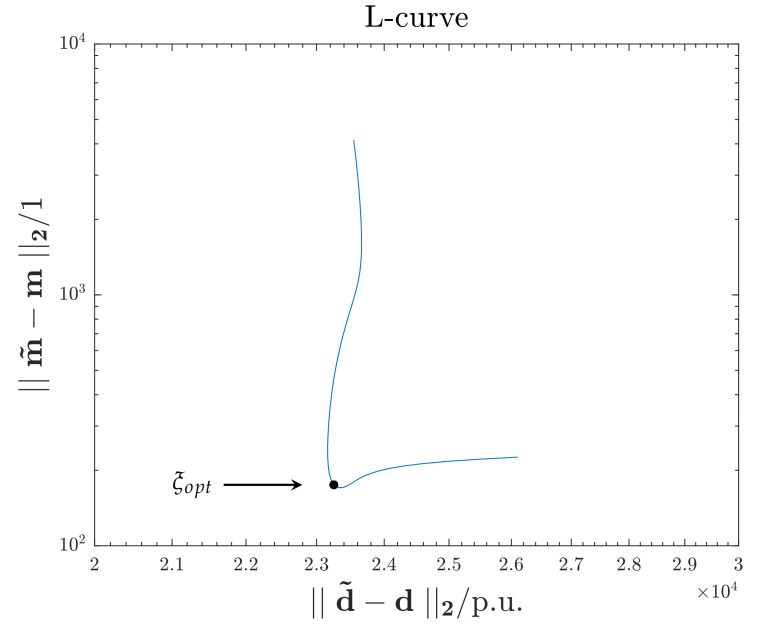
L-curve. The corner corresponds to estimated optimal value for the regularization parameter $\xi_{opt}=0.1$.

**Figure 13 sensors-21-00935-f013:**
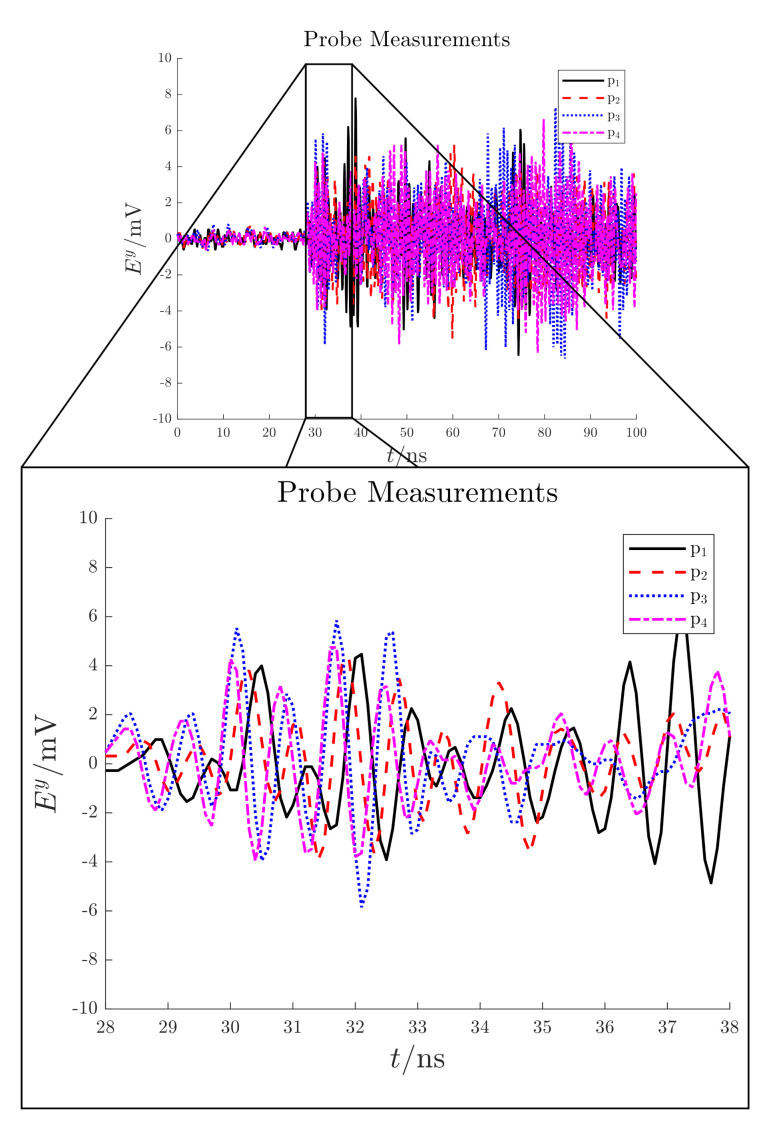
Test case 1. Electric field signals detected by the antennas in mV.

**Figure 14 sensors-21-00935-f014:**
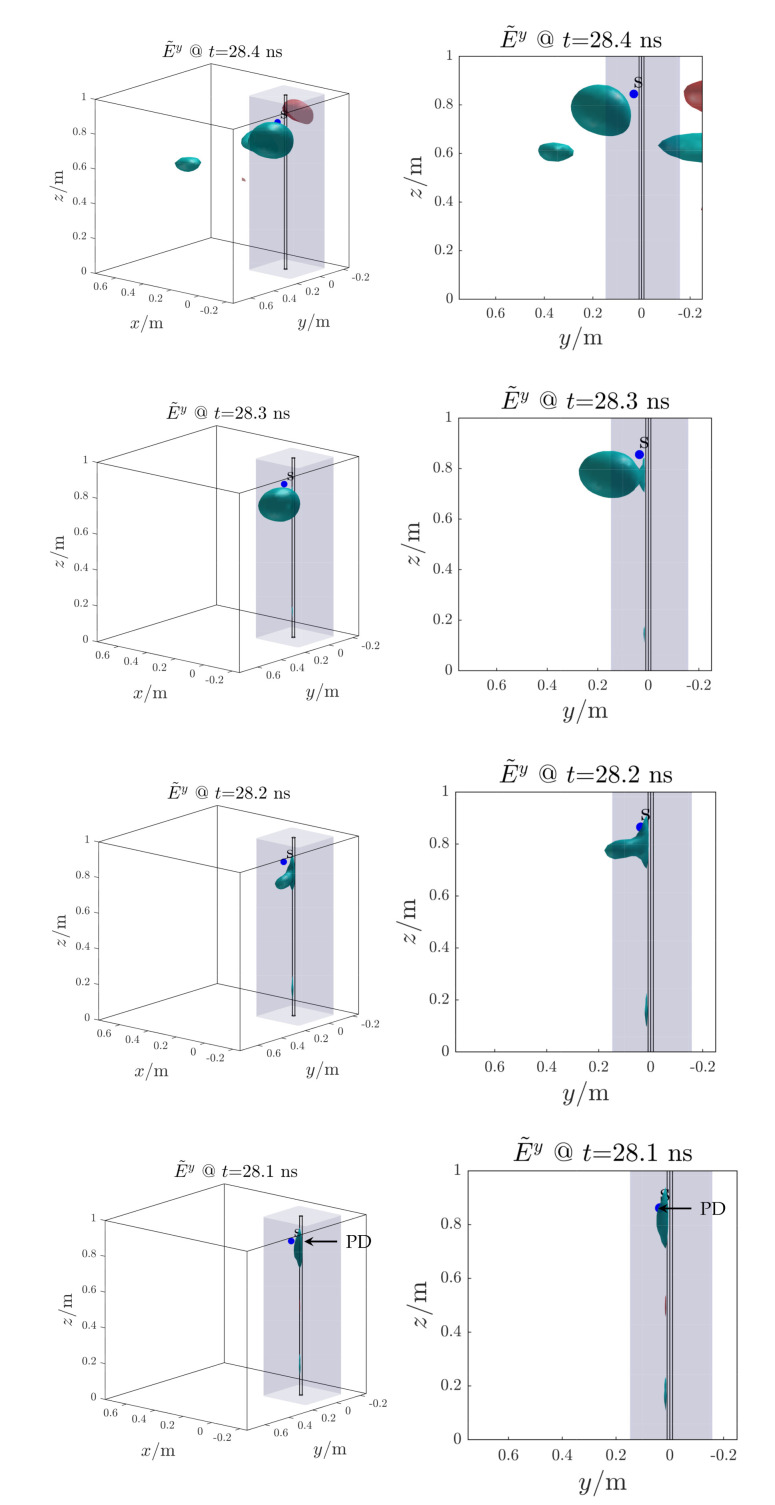
Test case 1. Reconstructed electric field distribution (propagation backward in time). The Partial Discharge (PD) source (*s*, blue dot) is correctly localized. The volume around the cable is blue highlighted.

**Figure 15 sensors-21-00935-f015:**
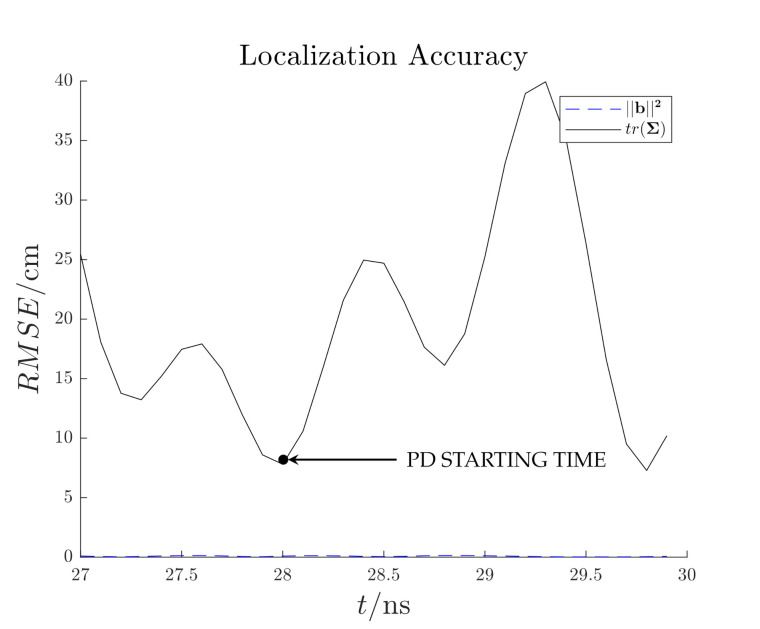
Test case 1. Estimated localization accuracy.

**Figure 16 sensors-21-00935-f016:**
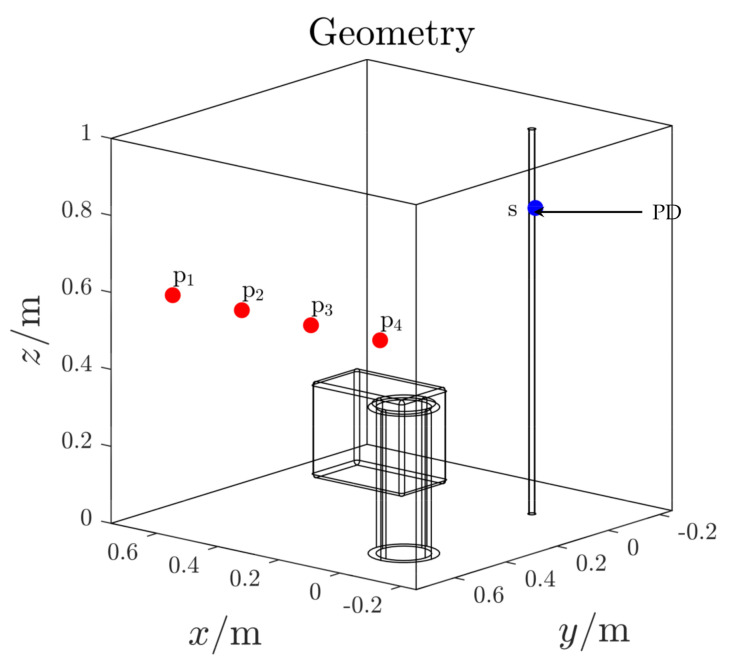
Test case 2. 3D domain including also the dielectric and ferromagnetic parts.

**Figure 17 sensors-21-00935-f017:**
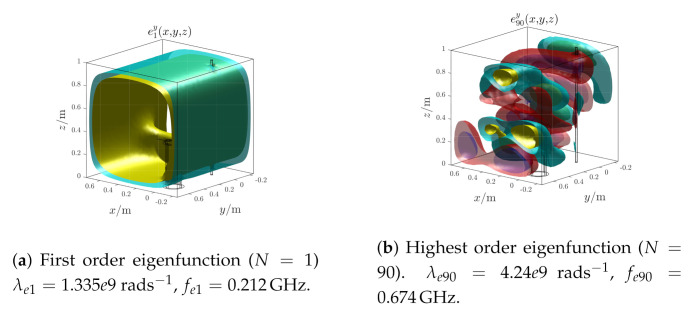
Test case 2. First and highest order electric field eigenfunctions.

**Figure 18 sensors-21-00935-f018:**
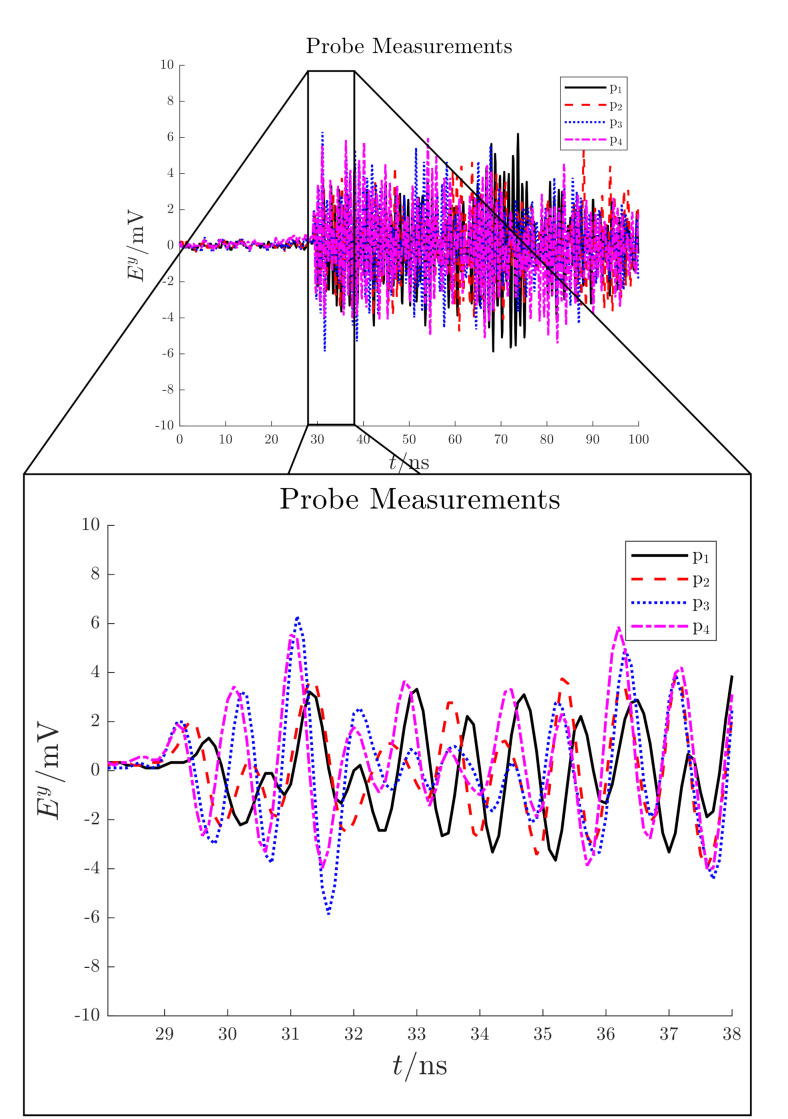
Test case 2. Electric field signals detected by the antennas in mV.

**Figure 19 sensors-21-00935-f019:**
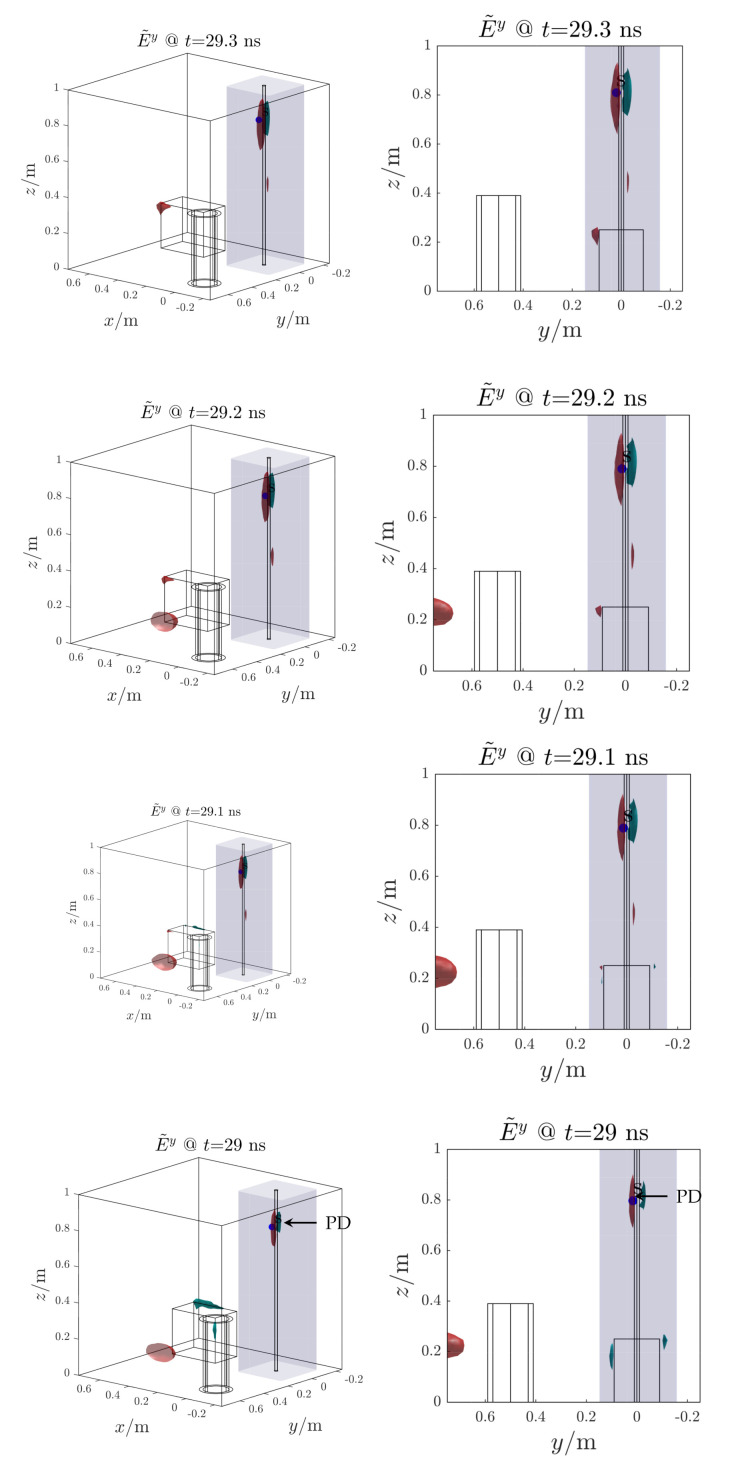
Test case 2. Reconstructed electric field distribution (propagation backward in time). The PD source (*s*, blue dot) is correctly localized. The volume around the cable is blue highlighted.

**Figure 20 sensors-21-00935-f020:**
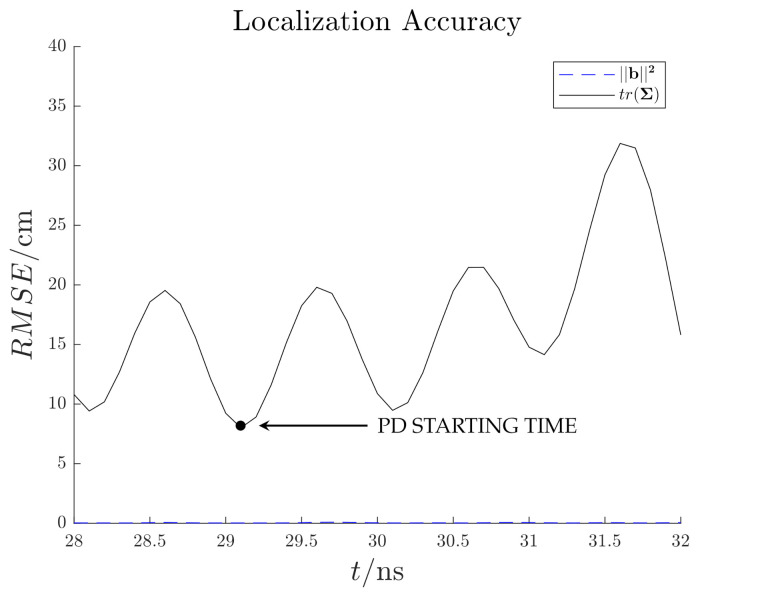
Test case 2. Estimated localization accuracy.

**Table 1 sensors-21-00935-t001:** Antenna locations.

Antenna	Location (x/m,y/m,z/m)
p${}_{1}$	(0.59,0.70,0.61)
p${}_{2}$	(0.36,0.70,0.61)
p${}_{3}$	(0.14,0.70,0.61)
p${}_{4}$	(−0.09,0.70,0.61)

## Data Availability

Not Applicable.

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
