# Peer review of "Experimental Results of Partial Discharge Localization in Bounded Domains"

_sensors, 2021, doi:10.3390/s21030935_

Round 1

Reviewer 1 Report

This work presents a novel diagnostic method to localize Partial Discharges (PDs) inside Medium Voltage (MV) and High Voltage (HV) equipment. The method is well suited for that equipment presenting a bounded domain with fixed Boundary Conditions (BCs). The reconstruction and localization tasks are based on the eigenfunctions series expansion method which intrinsically accounts for the physical information of the propagation phenomenon. This method is very novel and has been proven. However, validation is needed in more complex structures. Compared with Figure 11, the propagation process of electromagnetic wave in Figure 15 is not obvious, so it needs further analysis and explanation.

Author Response

We wish to thank reviewer #1 for the time spent in reading and reviewing our work and for the useful comments. We reported our response in the following points.

  • Figures 11 and 15 are 3D isosurfaces representing the electric wave propagation at arbitrarily chosen time instants. The algorithm explores all the estimated electric field values and it scales the max and min values. This visualization is useful for a human operator to interpret the results.

However, we would point out that the algorithm automatically provides all the results such as the reconstructed electric field distribution, PD location and RMSE trend. This automatized approach is more realistic, since human operators cannot always be present and may not be able to interpret the results.

  • In order to satisfy the reviewer request and to make it clear, for figure 15, the scale was manually changed and the previous plots are substituted.

Reviewer 2 Report

In this paper, new PD localization method is proposed. However, there are some unclear points. The author should revise the paper.

  1. Although the localization of PD is performed using four probes, the Correlation between the results and the proposed method is not discussed at all. The authors should revise the experimental results so that they can be a validation of the effectiveness of the proposed method. In addition, there is a lack of discussion on the validity of the results with four probes, and the usefulness of the method cannot be judged. The authors should show that the experimental conditions are sufficient for the validation of the proposed method.
  2. In Fig. 10, it is difficult to understand the difference of the waveform on each probes. The author should explain the expected result of this graph and emphasize it.
  3. In Fig 11, the red volume is undefined. The author should define the meaning of red volume.

Minor:

l.116 DAQ is already defined.

Author Response

We wish to thank reviewer #2 for the time spent in reading and reviewing our work and for the useful comments. We reported our response in the following points:

  • The main advantage of the proposed algorithm is that all the information (i.e., initial and boundary conditions and materials) about the physical phenomenon are encapsulated into the model. Using four antennas installed on one single surface and located according with a linear geometry, the algorithm is able to provide satisfactory results in terms of PD localization. We agree with the author that for real experimentation an optimal strategy is required to locate and to choose the antennas. This will be one of the future activities.

Moreover, in order to clarify the experimental conditions and to proof that they are carefully analyzed, two subsections are introduced "Noise Sources" and "Electromagnetic Model". In the first one, the background noise sources are analysed and measured; in the second, the experimental eigenfrequencies are measured to validate the assumptions on the adopted physical model.

  • In figure 10 a zoom was inserted to clarify the difference between the waveform on each antennas;
  • Figures 11 and 15 represent the electric wave propagation at arbitrarily chosen time instants. The blue and red volumes are 3D isosurfaces which values are chosen by the algorithm with respect to the min and max values of the reconstructed field. This visualization is useful for a human operator to interpret the results.

However, we would point out that the algorithm automatically provides all the results such as the reconstructed electric field distribution, PD location and RMSE trend. This automatized approach is more realistic, since human operators cannot always be present and may not be able to interpret the results.

  • Line 116 was corrected.

Reviewer 3 Report

Taking into account all the several features (technical aspects, quality and presentation), the accuracy, scientific quality, scientific content and interpretation of the results are interesting.

Technical

  1. The topic is appropriate for the journal;
  2. The work has a very clear structure;
  3. All the ideas are clearly expressed, and the concepts are understandable.
  4. The overall balance and structure of the paper is very good. Moreover, all the sections are necessary and properly written;
  5. English language seems to be appropriate.

Quality

The paper is focused on a novel diagnostic method to localize Partial Discharges inside 2 Medium Voltage (MV) and High Voltage (HV) equipment. The method proposed by the authors is  suitabel for that equipment presenting a bounded domain with fixed Boundary Conditions (BCs) such as Oil-Filled Power Transformers (OFPTs), Air Insulated Switchgears (AISs), Gas Insulated Switchgears (GISs) or 5 Gas Insulated Transmission Lines (GILs). It is based on Electromagnetic (EM) measurements which are used to reconstruct the EM field produced by the PD and localize the PD itself.

  1. The approach is interesting.
  2. It seems that the paper does not contain repetitions.
  3. The length of the work is appropriate and consistent with its scientific content.

Presentation

  1. The title is adequate and appropriate for the content of the article.
  2. The abstract contains information of the article.

I suggest to accept in present form 

Author Response

We wish to thank reviewer #3 for the time spent in reading and reviewing our work. We do appreciate the remarked features.

Round 2

Reviewer 2 Report

The authors revised adequately following the reviewer's comment.